# 3,5-Dinitrosalicylic Acid Adsorption Using Granulated and Powdered Activated Carbons

**DOI:** 10.3390/molecules26226918

**Published:** 2021-11-17

**Authors:** José A. Hernández, Laura Patiño-Saldivar, Alba Ardila, Mercedes Salazar-Hernández, Alfonso Talavera, Rosa Hernández-Soto

**Affiliations:** 1UPIIG, del Instituto Politécnico Nacional, Guanajuato 36275, Mexico; patinio2103@gmail.com (L.P.-S.); rohernandezs@ipn.mx (R.H.-S.); 2Politécnico Colombiano Jaime Isaza Cadavid, Medellín 4932, Colombia; anardila@elpoli.edu.co; 3Departamento de Ingeniería en Minas, Metalurgia y Geología, División de Ingenierías, Universidad de Guanajuato, Guanajuato 36025, Mexico; merce@ugto.mx; 4Unidad de Ciencias Químicas, Campus UAZ Siglo XXI, Universidad Autónoma de Zacatecas, Zacatecas 98160, Mexico; talavera@uaz.edu.mx

**Keywords:** DNS, nitroaromatic compounds, removal, equilibrium, mass transfer

## Abstract

Some nitroaromatic compounds are found in wastewater from industries such as the weapons industry or the wine industry. One of these compounds is 3,5-dinitrosalicylic acid (DNS), widely used in various tests and frequently found as an emerging pollutant in wastewater and to which the required attention has not been given, even though it may cause serious diseases due to its high toxicity. This study investigated the adsorption of DNS using granulated activated carbon (GAC) and powdered activated carbon (PAC) at different temperatures. The results show that in equilibrium, the adsorption takes place in more than one layer and is favorable for the removal of DNS in both GAC and PAC; The maximum adsorption capacity was obtained at 45 °C, with values of 6.97 mg/g and 11.57 mg/g, respectively. The process is spontaneous and exothermic. In addition, there was a greater disorder in the solid-liquid interface during the desorption process. The predominant kinetics using GAC (7.14 mg/g) as an adsorbent is Elovich, indicating that there are heterogeneous active sites, and when PAC (10.72 mg/g) is used, Pseudo-second order kinetics predominate, requiring two active sites for DNS removal. External mass transfer limitations are only significant in GAC, and ATR-FTIR studies in PAC demonstrated the participation of functional groups present on the adsorbent surface for DNS adsorption.

## 1. Introduction

Water pollution has been a major problem worldwide for many years, causing serious health consequences for all living beings [1,2,3]. This is due to the fact that several industrial processes produce large amounts of wastewater containing pesticides, pharmaceuticals, and heavy metals, among others [4,5,6]; In addition, other anthropogenic activities, such as urban or agricultural, together with the widely reported droughts of recent years, have caused a global water shortage [1], which is reflected in that more than 40% of the world’s population does not have access to drinking water [7]. These circumstances have led to a search for better use of water and to focus on the reuse/recycling of wastewater as a viable option to alleviate the problem of water shortages, since, although it cannot be properly used for human consumption, wastewater properly treated can be used for processes that do not require high-quality standards [1,8,9]. To achieve the reuse of water from contaminated effluents, various treatment methods have been proposed, since due to the high costs generated by reuse treatments and the increasingly stringent sanitary regulations that must be met, it is necessary to seek options that are technologically and economically feasible [3,9]. Some of the treatment methods implemented are oxidative processes [10,11], reverse osmosis [1], photodegradation [11], ultrafiltration [1], sedimentation [5], ozonization [12] and ion exchange [13,14]. Another treatment alternative for the adsorption of pollutants is the use of activated carbon (AC), which has shown stability and high adsorption capacity, easy handling, low energy consumption, and versatility, and they do not form reaction by-products [1,15,16]. AC is widely used in water treatment [1,6,11,12,13,17] because it has properties of interest such as high contact surface, porosity, and abundant surface groups [12] that are more stable compared to those that are found in clays, zeolites, and graphene-based adsorbents [11].

In several works, the obtaining of coal from agro-industrial waste has been studied, which is usually economical and friendly to the environment; This type of carbon is characterized by having textural and chemical properties that allow it to have good efficiency compared to commercial adsorbents [1,2,15,18,19,20]; however, because the chemical and physical properties of carbon, such as specific surface area and pore distribution, among others, are directly related to the source and conditions of carbon synthesis that are used, the use of these materials is limited as an alternative in the processes for obtaining adsorbents [1,18,21]. For this reason, granulated activated carbon (GAC) continues to be used for the treatment of polluted water; This material is effective in obtaining drinking water from contaminated effluents from different industries, which makes it a technical-economic perspective [10]. Nitroaromatic compounds are highly toxic and probably carcinogenic substances for living beings, and are highly resistant to natural biodegradation, and are thus a serious threat to the environment [15] due to factors such as low affinity for water and high uptake of electrons due to the nitro groups (NO_2_) that are next to the aromatic ring. Its concentration in wastewater increases every year due to the different activities of human beings [1,2,15]. The nitroaromatic compound 3, 5-dinitrosalicylic acid (DNS), is used as a quantifier for reducing sugars in products of the sugar industry, such as wines, etc., it is an organic aromatic compound with physicochemical properties similar to other nitroaromatic compounds and causes various ailments such as irritation to the intestinal tract, metabolic disorders, and kidney damage. Since there are no studies regarding the removal treatment of adsorbent materials of this type of compound in wastewater [15,21,22,23], the present work aims to study the adsorption of DNS in equilibrium and dynamically using GAC and powdered activated carbon (PAC) at different temperatures in order to determine if the process has limitations due to external and intraparticle mass transfer.

## 2. Results and Discussion

### 2.1. DNS Adsorption Isotherms

The equilibrium adsorption analysis allows us to obtain the parameters of the different isotherm models (Table 1), indicating what type of process is carried out in the removal of pollutants. 

In Figure 1, the adsorption capacity of DNS at different temperatures using GAC (Figure 1a) and PAC (Figure 1b) is shown. It is also observed that the adsorption of DNS in both adsorbents is significantly affected with the increase in temperature, presenting greater adsorption when increasing the temperature; thus at 45 °C, the maximum adsorption capacity of 6.97 and 11.57 mg/g was obtained for GAC and PAC, respectively. This may be due to the environment of the removal process, which can cause the adsorbent surface to increase its participation in DNS adsorption with increasing temperature [21,25,26]. Similar behavior has been reported in the literature using commercial AC [21]; in other studies, using this adsorbent it has also been observed that there is an increase in the adsorption capacity of different compounds based on phenols [27], perchlorates [25], and heavy metals [26], among others [12].

Table 2 shows the different parameters obtained from the isotherm models that were used to adjust the experimental data. Taking the coefficient of determination (R^2^) and the normalized standard deviation (Δq%) as criteria to choose the best fit, it was found that regardless of the temperature for both adsorbents, the best fit is that of SIPS, which indicates that the adsorption is performed on the surface in more than one layer; Furthermore, when determining the parameters of this model, the value of n_s_ is greater than 1, which implies that adsorption is a physical process [28,29,30]. This result is also obtained with the Freundlich model, since n > 1 implies that the adsorption of DNS in GAC and PAC takes place on the surface of the adsorbents [24,28,31,32]. Additionally, the adsorption energy (E) obtained with the DR model is less than 8 kJ/mol, confirming that physical adsorption is the mechanism with which DNS is adsorbed in AC. Also, it can be observed that this energy increases with the decrease in temperature for both adsorbents, which allows us to mention that less energy is required to remove this nitroaromatic compound from the solution when the temperature increases [32,33]. Langmuir’s model indicates that the removal of this pollutant is a favorable process (0 < R_L_ <1), this is directly related to the shape of the isotherms [28,34,35] presented in Figure 1. 

The study of the removal of sulfosalicylic acid and DNS, using GAC treated with commercial oxidative methods at different temperatures, found that the best fit model is that of Langmuir [21], presenting a great adsorption capacity. In some pharmaceutical compounds, such as diclofenac, acetylsalicylic acid, caffeine, etc., the model of best fit is that of Toht [36], although, in other pharmaceutical compounds such as ibuprofen, naproxen, among others, the best model was that of Langmuir [11]. Also, the adsorption of phenols and derivative compounds has been carried out in GAC and other low-cost adsorbents.

It was observed that the best models for these types of compounds are those of Freundlich and Langmuir [27]. With the use of GAC treated with HCl and KOH, a great adsorption capacity was obtained in an equilibrium of perchlorate [25] and trifluoromethane (CHF_3_) [37], where they found that for perchlorate ions the best model was that of Freundlich, while for CHF_3_ the best was Langmuir. On the other hand, for the adsorption of primidone in PAC, it was found that in equilibrium the Freundlich model is the one that best fits for the experimental data [12]. In the adsorption of heavy metals such as Sr (II) and Ba (II) there is also a great adsorption capacity using AC, showing that the best model that fits the experimental data is that of Langmuir, although this same model has been used to fit isotherm data of methylene blue adsorption using mesopore carbon obtained from PET [19]. In most studies, up to three isotherm models are used to fit their obtained experimental data, in contrast to what was done in the present work, where six different models are used to analyze the experimental data to fully explain the process of adsorption of DNS and other emerging compounds such as pharmaceutical residues in GAC and PAC, since each of these different models provides valuable and complementary information on the process.

The removal percentage of DNS using different temperatures is shown in Figure 2 for both samples of AC. In this figure, it could be observed that the increase in DNS removal had a linear trend concerning temperature, having 87 and 99% removal for GAC and PAC, respectively. In addition, it was noted that at 45 °C there was the maximum point of this percentage in GAC, but at 25 °C there was the minimum removal value for PAC, which indicates that it may have a direct relationship between the surface of the adsorbent, the operating temperature and the environment in which the DNS clearing process takes place. In the case of primidone removal, a removal percentage of 100% was obtained at 25 °C [12], and in the case of removal of chemical oxygen demand (COD), dissolved organic carbon (DOC) and total hydrocarbons of petroleum (TPH) present in ship wastewater, a removal percentage of 80, 68 and 76%, respectively, was shown using GAC [13].

### 2.2. DNS Adsorption Thermodynamics

The study of the DNS removal process was determined from the use of the thermodynamic parameters found with the adjustment of the experimental data using the different isotherm models, and the values of the parameters obtained are shown in Table 3. From these data, it can be observed that for both samples the process is spontaneous (ΔG < 0) and that the Gibbs free energy increases with increasing temperature; this change is more noticeable for the GAC adsorbent while for PAC the increase is gradual. DNS adsorption has an endothermic nature (ΔH > 0), which indicates that as the temperature increases, the adsorption capacity of GAC and PAC increases, which suggests that the electrostatic forces that are established on the surface of the adsorbent intervene significantly in the process [21]; on the other hand, with the change in entropy (ΔS > 0) the randomness in the solid-fluid interface increases during the adsorption process, which is expected behavior for adsorption processes in the liquid phase [12,14,25,26,38]. 

### 2.3. DNS Adsorption Kinetic Study

In Figure 3, the effect of the variation of adsorbent concentration (C_ads_) on the adsorption of DNS is shown. It can be observed that the adsorption capacity in GAC and PAC decreases almost linearly with the increase in C_ads_, which varies from 2 to 10 g/L. This causes the removal percentage to change from 67% at 25 °C to 48 and 53% at 35 and 45 °C, respectively, for GAC. The same variation of C_ads_ was used in PAC where the removal percentage at 25 °C was 69.8% and increased to 90 and 91% at 35 and 45 °C, respectively. This behavior is due to the active adsorption sites that have been exhausted or blocked by the variety on the surface of both adsorbents. In addition, the increase in the removal percentage can be assumed by the reduction of the adsorbate-adsorbent ratio with the increase of the adsorbent concentration [12,28,39]. DNS removal using GAC and PAC also shows an increase with temperature, obtaining a removal percentage of 64, 74.6, and 86.6% at 25, 35, and 45 °C, respectively, for GAC. While for PAC the removal percentages of 59.3, 91, and 92.1% were obtained at 25, 35, and 45 °C, respectively. This increase in removal may be due to the concentration of the adsorbents, where the availability of the active adsorption site is more accessible. With these results, it can be mentioned that PAC retains more of the DNS on its surface with increasing temperature [12].

The kinetic study of the DNS removal process allows us to obtain the mechanism that controls the adsorption (Table 4) because one of the fundamental factors is to know if there are mass transfer problems. 

Table 5 shows the parameters obtained from the adjustments of the experimental data of the adsorption of DNS with GAC at the different temperatures and taking R^2^ and Δq% as criteria to choose the best model (resulting in Elovich), from which it is inferred that on the surface of GAC, different adsorption energies are used for the capture of the DNS molecule on its surface. In this study, the DI and DE models indicate whether the removal process may have limitations due to internal and external mass transfer, which significantly affect the removal of contaminants present in the solution. In the case of DNS adsorption with GAC, although these models do not present the best fit of the experimental data, the R^2^ and Δq% values show that it can interfere in the DNS elimination process, for which an Analysis of the influence of the intraparticle and external mass transfer for this adsorbent was made. 

In the case of PAC (Table 6), the best model that fits the experimental data is that of PSO, which shows that the adsorption of DNS in the AC powder requires two active sites for each molecule of DNS to carry out the removal of this nitroaromatic contaminant [12,21]. For this adsorbent, it can be mentioned that based on the values of R^2^ and Δq% obtained, there are no limitations due to external or intraparticle mass transfer in such a way that the removal process is carried out on the surface of the material and the functional groups that are in it, which partially explains why there is better adsorption using the powdered adsorbent in comparison to its granular form. 

Previous studies using GAC treated with different acids [25] to remove perchlorate found that the best kinetic model was the pseudo-second order. Similarly, in the removal of different emerging pollutants such as caffeine, gallic acid, and ibuprofen, among others [36], it was observed that the best fit is obtained with the pseudo-second order model. On the other hand, in the adsorption of primidone using PAC, the experimental data of the removal of this compound are also fitted through the pseudo second order model. These results agree with what was obtained in this study, being the same model with which the experimental data of DNS removal in PAC are adjusted [12]; however, in the case of GAC, the result differs from that obtained in the present study. Nevertheless, this could be derived from the fact that in the aforementioned studies only two kinetic models were used to fit the experimental data, which could limit the analysis of the pollutant adsorption process [25,36]. 

### 2.4. Mass Transfer Study on DNS Adsorption in GAC and PAC

The analysis of the external and intraparticle mass transfer process can have a significant effect on the adsorption of DNS, since the information obtained in Table 5 and Table 6 indicate that the limitations due to transport of matter could be present in the removal of DNS, which makes it necessary to carry out an analysis of this phenomenon [36]. Figure 4a is constructed using the external diffusion model as a basis, where it can be seen that for GAC there is a straight line at the experimental points for up to 6 h of contact in the adsorption process, which indicates the possibility that there is control by external mass transfer in the process at any temperature [28]; this type of behavior is similar to that reported in the literature for the adsorption of organic matter components from ship waste [13]. Figure 4b shows the representation of the experimental data obtained for PAC in the adsorption of DNS, where a straight line was observed at 1 h of contact with the adsorbent. This indicates that there are no external mass transfer problems using particles of AC, because DNS is quickly removed from the solution diffusing towards the surface. Furthermore, this suggests that the adsorption process in AC particles may be due to the functional groups present on the surface. Based on these results, it is established that it is important to know if there are also internal mass transfer problems and thus be able to determine if the DNS removal process is significantly affected by the presence of this phenomenon.

Figure 5 is constructed by comparing q vs t^0.5^, which refers to the intraparticle mass transfer model [14,28]. In Figure 5a, only two stages can be seen: external diffusion and equilibrium, since the intraparticle diffusion stage is not clearly observed [22,27,28,36]. This indicates that the external mass transfer is directly involved in the adsorption of DNS, which causes the DNS molecules to delay being removed from the solution and to be captured on the surface of the GAC, therefore It needs more adsorption energy, making the adsorbent capacity lower compared to powdered AC, which is in agree with what was obtained in the study of adsorption in equilibrium in this study.

The analysis of limitations by intraparticle mass transfer in the adsorption with PAC (Figure 5b) indicates that two stages can be observed, external diffusion, which is a short stage, since it only occurs in the first hour of the process, giving kinetic constants of an order of magnitude much higher than those obtained by GAC. Later, the equilibrium stage is noted. This result shows that the adsorption process does not have external and intraparticle mass transfer limitations, but that the process is controlled by the capture of the molecules on the surface of the carbon particles.

This study showed that one of the causes for which there is a greater adsorption capacity and percentage of removal of DNS is the use of powdered AC, since it only depends on the participation of the functional groups that are found on the surface; conversely, for GAC it not only depends on the participation of the adsorbent surface but is limited by the effect of external mass transport, confirming that more energy must be used to remove the DNS from the solution compared to PAC.

### 2.5. Characterization of PAC

In the FTIR spectra of PAC before and after the adsorption of DNS shown in Figure 6, different bands are observed; the band at 3432 cm^−1^ is attributed to the vibration of the extension of the hydroxyl groups (O-H), and at the adsorption of water on the surface of the adsorbent, the band at 1738 cm^−1^ is attributed to the extension of the C=O groups present in the carboxylic acid or aldehydes, while the band at 1570 cm^−1^ is attributed to the vibrational stretching of the C=C and C=O groups of the aromatic rings, the bands at 1191 and 1043 cm^−1^ refer to the C-O and O-H bond of the phenolic compounds on the surface of PAC [20,25,26,29,42,43,44]. These functional groups are those that allow the DNS molecules to be captured, since when observing the different bands, the increase of intensity allows us to infer that the capture of the molecules on the surface of the adsorbent is due to physical attraction. However, in the case of GAC having the same functional groups, DNS molecules delay interaction with the surface due to their external mass transfer limitations. This result confirms that it is necessary to use AC in powder so that there is a greater removal of pollutants in the shortest possible time, allowing a more specific interaction between the surface of the carbon particles with the molecules of the different pollutants that may be in sewage.

Table 7 shows the textural properties provided by the company NO-RIT [45] together with the isoelectric point and the elemental analysis of the adsorbent, where it can be observed that the value of the isoelectric point gives us an environment on the surface where it can be inferred that the electrostatic forces are acting in addition to the functional groups present on the surface of PAC, which implies that in the physical adsorption process found in the quantitative studies and in equilibrium, the attractive forces also intervene. Furthermore, the textural properties of the adsorbent allows for a large surface area to avoid the fact that the DNS molecules agglomerate in only part of the material surface, allowing more than a single monolayer to form [46].

The TGA results show us that there is a humidity in the material of 16%, with the presence of ashes (12.5%) and volatile material in general (8.63); this is characteristic of this material [44] and allows us to have a PAC effective area of more than 70% corresponding to fixed carbon. The following table (Table 8) shows the results of the analysis of the acid groups on the surface of the PAC, where it is shown that the groups determined by the Boehm method agree with the FTIR results, these being the phenolic groups the ones with the highest presence which implies that the surface is weak acid as indicated by the value of the isoelectric point [45,46]. Based on these results obtained during this work, it can be noted that it is possible to eliminate the DNS residues that are used in different processes to know the reducing sugars of common consumer products such as wines, etc., and that a better adsorption capacity can be obtained when mass transfer limitations are not significant for the adsorption of this pollutant using activated carbon particles compared to activated carbon granules. Furthermore, having a removal of almost 100% is very similar to that obtained in other studies [15,21]. Although the adsorption capacity is low, it is possible to reduce the concentration of 3, 5-dinitrosalicylic acid in solution.

## 3. Materials and Methods

### 3.1. Reagents

All reagents used were analytical grade, 3, 5-dinitrosalicylic acid (DNS; CAS No. 609-99-4; Sigma Aldrich, St. Louis, MO, USA) and Xylose (CAS No. 58-86-6). General Purpose Granulated Activated Carbon (DARCO 20 × 40) was purchased from Norit Americas Inc., Marshall, TX, USA, [46], part of GAC was pulverized with a ball mill (Gunt Hamburg CE-245), Barsbüttel, Germany and sieved (Ro-Tap RX-30, Mentor, OH, USA) to a particle size of 100 mesh (0.297 mm). 

### 3.2. Characterization of AC

Attenuated Total Reflectance-Fourier transform spectroscopy (ATR-FTIR) analyses before and after adsorption of DNS were carried out over the wave number range of 4000–400 cm^−1^ using a Thermo Scientific Nicolet iS10 analyzer, and 32 scans were obtained with a resolution of 4 cm^−1^. The determination of the isoelectric point of OP was carried out using the mass titration method described by Hernandez et al. [14], and the textural properties (surface area, diameter and pore volume) were provided by Norit Americas Inc. [44]. The thermogravimetric analysis (TGA) of nDCPD was performed in a scale (TGA Q500 TA Instruments, New Castke, Delaware, USA) placing 20 mg of sample in an aluminum basket. The sample was heated starting at room temperature reaching 700 °C with a heating rate of 10 °C/min [14]. Additionally, the elemental analysis of the samples was analyzed in duplicate in a CHNSO (LECO model Truspec micro, Geleen, The Netherlands), using the ASTM D-5373-08 method. The CHN analysis was performed at 1050 °C in a helium environment and sulfur at 1350 °C in an oxygen environment. The analysis of acidic and basic sites was achieved by Boehm’s method [45].

### 3.3. DNS Adsorption at Equilibrium

For the equilibrium studies of DNS adsorption in GAC and PAC, 0.25 g of bioadsorbent with 50 mL of DNS solution were placed in a shaker (ZICHENG ZHWY-200D, Shanghai, China) with an orbital shaking of 200 rpm at 25, 35, and 40 °C for 24 h of contact time, varying the concentration between 0 and 1.5 mg of Xylose/mL. After this time, a sample aliquot was taken, which was centrifuged (HERMLE Z383K, Riga, Latvia) at 4000 rpm for 10 min, and, subsequently, the absorbance in UV-Vis spectrum was measured (Jenway 6705, Waltham, Massachusetts, USA) [30,40]. For the quantification of the DNS, Xylose was used as a reference, constructing a calibration curve and measuring the absorbance at 550 nm. The adsorption capacity (q) of DNS can be determined with the following equation [41]:(1)$$q_{e}=\frac{\left(C_{0}\;-\;C\right)V}{m}$$
where C_0_ and C correspond to the initial and equilibrium concentration (mg L^−1^) respectively, V is the volume of solution (L) and m is the mass of the adsorbent (g). The DR isotherm (Table 1) is generally applied to express the adsorption mechanism with a Gaussian energy distribution on a heterogeneous surface. The mean energy of sorption, E, is calculated by Equation (2) [24,35]:(2)$$E=\frac{1}{\sqrt{{2k}_{\mathrm{DR}}}}$$

The magnitude of E is useful for estimating the type of sorption reaction [10]. The Gibbs free energy (ΔG), which allows knowing the spontaneity of the adsorption process, was calculated by using Equation (3) [41,42]:
∆G = −RTln55.5 K_L_(3)
where K_L_ is the Langmuir model constant (L/mol), R is the ideal gases constant (kJ/mol K) and T is the absolute temperature (K). The separation factor, R_L_, which allowed predicting the affinity between adsorbent and adsorbate, was calculated using the equation [26]:(4)$$R_{L}=\frac{1}{1+K_{L}C_{0}}$$

Of all these models, it has to be chosen the one which is the best fit, for these two different criteria were used: the first is the deterministic coefficient (R^2^) which must have a value close to 1 to be considered as a possible fit and the second is the normalized coefficient of determination, Δq, which was determined using Equation (5) [40,43]:(5)$$∆q=\sqrt{\frac{{\left(\frac{q_{\exp}{\;-\;q}_{\mathrm{cal}}}{q_{\exp}}\right)}^{2}}{N-1}}*100\;$$
where N is the number of data, q_exp_ and q (mg/g) are the values of the experimental adsorption capacity calculated by the model, respectively. Another parameter to determine is the removal percentage, %R_DNS_, which was determined with Equation (6) [40,42]:(6)$${\%R}_{\mathrm{DNS}}=\frac{\left(C_{0}-C\right)}{C_{0}}*100$$

### 3.4. DNS Batch Adsorption Kinetics

The DNS removal kinetics were carried out in batches to observe the evolution of GAC and PAC adsorption. The mass varied from 0.1 to 0.5 g with 50 mL of solution at a concentration of 1.45 mg Xylose/mL, and the closed containers were placed in a shaker (ZICHENG ZHWY-200D, Shanghai, China) at a speed of 200 rpm and 25, 35, and 45 °C. Aliquots were taken every 1 h to separate the bioadsorbent by centrifugation (HERMLE Z383K, Riga, Latvia) at 10,000 rpm, and the liquid supernatant was analyzed by UV-Vis spectroscopy (Jenway 6705, Waltham, MA, USA) [24,40,45]. 

## 4. Conclusions

In the present study, the ability to remove DNS was analyzed using commercial AC in granular and powdered form, and the removal of DNS was evaluated in a kinetic process and at equilibrium. In the equilibrium study, it was revealed that the adsorption process is carried out on the surface of the material without the limitation of having only a monolayer of interaction between the adsorbate and the adsorbent; this is likely attributable to the fact that the process is favored by the increase in temperature due to the endothermic nature of DNS adsorption.

For its part, the kinetic study showed that GAC and PAC have different removal mechanisms depending on the shape of the material, and that specifically for GAC there is a surface with heterogeneous adsorption energies, in addition to presenting external mass transfer problems. On the other hand, in PAC it was found that two active sites are needed to adsorb a DNS molecule, and it showed that there are no external mass transfer problems that make a big difference in its adsorption capacity. The results obtained suggest that PAC has a better DNS adsorption capacity compared to GAC, which is likely a consequence of the difference in the characteristics of the shape of the adsorbent material. In the ATR-FTIR analysis, the different functional groups that participate in the physical adsorption of pollutants such as DNS were located. It was shown that the surface of the adsorbents is acidic in nature, which allows us to infer that physical adsorption also occurs with the participation of electrostatic forces for the DNS removal.

## Figures and Tables

**Figure 1 molecules-26-06918-f001:**
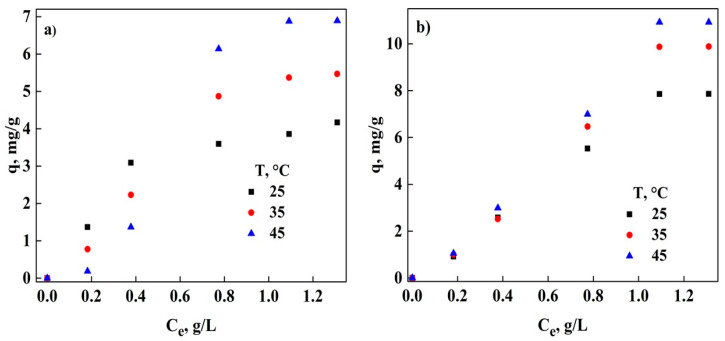
Evolution of DNS adsorption at different temperatures; (**a**) GAC and (**b**) PAC.

**Figure 2 molecules-26-06918-f002:**
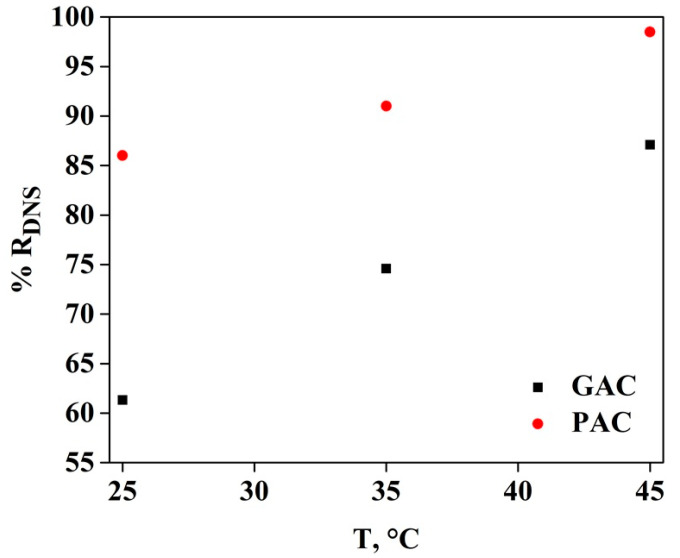
Influence of temperature on the percentage of removal of DNS in AC.

**Figure 3 molecules-26-06918-f003:**
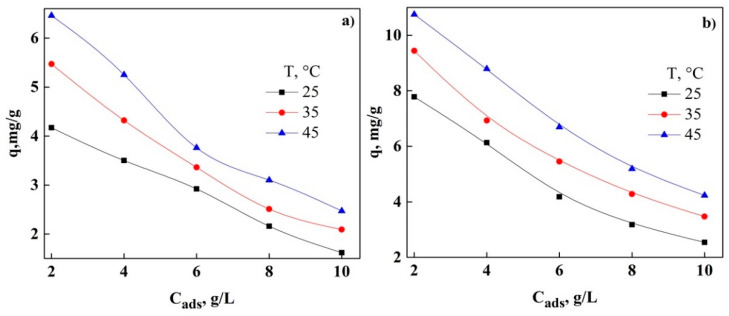
Effect of adsorbent concentration on DNS adsorption: (**a**) GAC and (**b**) PAC.

**Figure 4 molecules-26-06918-f004:**
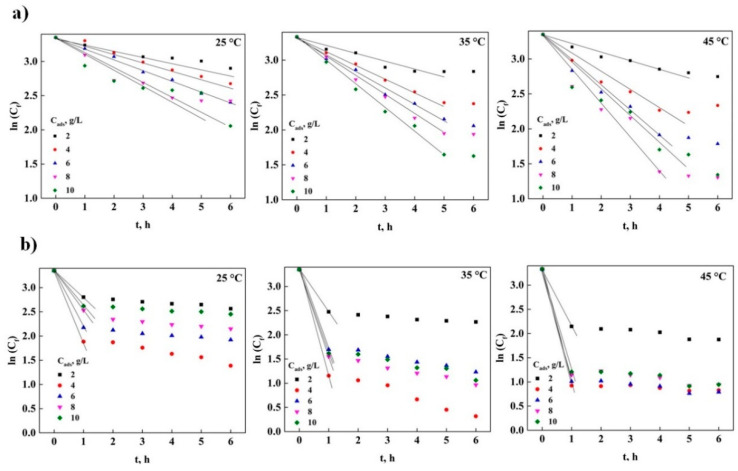
External mass transfer analysis at different temperatures: (**a**) GAC and (**b**) PAC.

**Figure 5 molecules-26-06918-f005:**
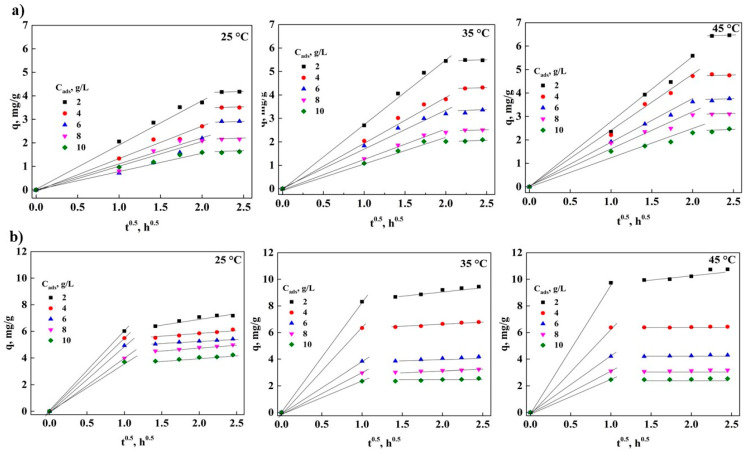
External mass transfer analysis at different temperatures: (**a**) GAC and (**b**) PAC.

**Figure 6 molecules-26-06918-f006:**
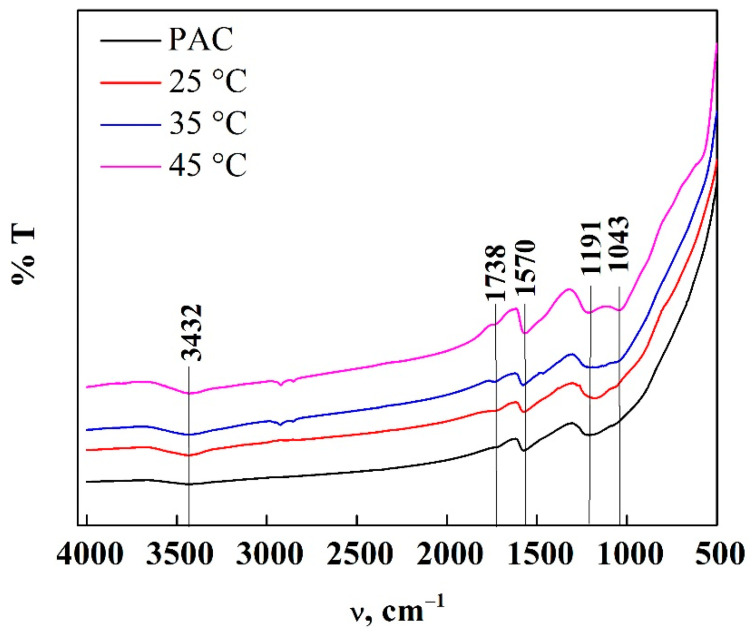
ATR-FTIR spectra of PAC before and after adsorption.

**Table 1 molecules-26-06918-t001:** Non-linear adsorption isotherm models [24].

Model	Equation
**SIPS**	$q_{e}=\frac{q_{m}{\left(K_{s}C_{e}\right)}^{n_{s}}}{1+{\left(K_{s}C_{e}\right)}^{n_{s}}}$
**Redlich-Peterson (RP)**	$q_{e}=\frac{K_{R}C_{e}}{{1+a}_{R}C_{e}^{\beta }}$
**Langmuir**	$q_{e}=\frac{q_{m}K_{L}C_{e}}{{1+K}_{L}C_{e}}$
**Temkin**	$q_{e}=A+\mathrm{Bln}\left(C_{e}\right)$
**Freundlich**	qe=KFCe1n
**Dubinin-Radushkevich (DR)**	$q_{e}{=q}_{m}\exp\left({-k}_{\mathrm{DR}}\varepsilon^{2}\right)$

**Table 2 molecules-26-06918-t002:** Adjustment of the isotherm models in the adsorption of DNS in AC.

Models	Parameters					
	GAC			PAC		
	25 °C	35 °C	45 °C	25 °C	35 °C	45 °C
**Langmuir**						
K_L_, L/mg	0.345	0.8	2.77	0.141	0.207	0.403
q_m_, mg/g	5.238	10.9	22.5	22.6	40.9	69.26
R_L_	0.19–0.67	0.46–0.87	0.66–0.94	0.62–0.93	0.62–0.93	0.83–0.98
R^2^	0.976	0.958	0.897	0.968	0.958	0.958
Δq, %	11.4	35.4	44.4	8.36	14.1	23.9
**Freundlich**						
K_F_ *	3.42	4.68	5.6	6.31	7.66	8.361
n	2.58	1.5	1.16	1.25	1.123	1.031
R^2^	0.95	0.932	0.88	0.956	0.95	0.95
Δq, %	1.79	51.96	10.2	6.65	4.88	5.87
**Temkin**						
A, L/mg	1.25	2.45	3.67	3.61	4.69	5.163
B, kJ/mol	21.4	7.52	5.24	6.41	5.79	5.822
R^2^	0.972	0.977	0.945	0.977	0.961	0.96
Δq, %	61.1	9.73	6.73	9.48	12.9	14.0
**Sips**						
K_S_, L/mg	4.17	2.29	1.97	1.64	1.49	1.45
q_m_, mg/g	4.21	5.84	6.97	8.31	9.82	11.6
n_S_	2.287	2.54	4.77	2.02	2.27	2.14
R^2^	0.994	0.997	0.999	0.99	0.986	0.985
Δq, %	0.39	2.98	0.504	2.52	0.26	2.65
**RP ****						
K_R_, L/g	14.5	8.71	7.7717	9.0974	9.6986	10.7489
a_R_, (L/mg)β	2.768	0.80	0.3448	0.4033	0.2368	0.2409
β	1.00	1.00	1.00	1.00	1.00	1.00
R^2^	0.976	0.958	0.897	0.967	0.958	0.96
Δq, %	0.52	3.91	5.06	3.55	3.77	3.67
**DR *****						
q_m_, mg/g	4.49	6.79	9.65	10.03	13.3	16.63
k_DR_, J^2^/mol^2^	0.048	0.099	0.15	0.127	0.14	0.179
E, kJ/mol	3.23	2.25	1.84	1.983	1.86	1.67
R^2^	0.961	0.98	0.95	0.982	0.975	0.951
Δq, %	3.42	10.77	17.9	12.3	15.4	23.4

* K_F_, [(mg/g)(L/mg)]^1/n^, ** Redlich-Peterson (RP), *** Dubinin-Radushkevich (DR).

**Table 3 molecules-26-06918-t003:** Thermodynamics of DNS adsorption in GAC and PAC.

T, °C	−ΔG, kJ/mol	ΔH, kJ/mol	ΔS, kJ/mol K
**GAC**			
25	42.03	82.29	0.398
35	40.26		
45	39.44		
**PAC**			
25	37.26	41.46	0.331
35	37.14		
45	36.97		

**Table 4 molecules-26-06918-t004:** Kinetic models of adsorption [40,41].

Model	Equation
Pseudofirst order (PFO)	$q\;=q_{\max}\left[1-\exp\left(-k_{1}t\right)\right]$
Pseudo second order (PSO)	$q=\frac{t}{\frac{1}{k_{2}q_{\max}^{2}}+\frac{t}{q_{\max}}}$
Elovich	$q=\frac{1}{\beta }\ln\left(\alpha \beta \right)+\frac{1}{\beta }\mathrm{lnt}$
Intraparticle Diffusion (ID)	${q=k}_{\mathrm{id}}t^{0.5}$
External Diffusion (ED)	$q=\frac{C_{0}V}{m}\left[1-\exp\left({-k}_{\exp}t\right)\right]$

**Table 5 molecules-26-06918-t005:** Kinetic constant of DNS adsorption patterns in GAC.

Model	2 g/L			4 g/L			6 g/L			8 g/L			10 g/L		
PFO	25 °C	35 °C	45 °C	25 °C	35 °C	45 °C	25 °C	35 °C	45 °C	25 °C	35 °C	45 °C	25 °C	35 °C	45 °C
q_max_	4.51	6.2	7.14	9.06	4.23	4.98	6.64	3.20	3.92	2.31	2.67	3.83	1.61	2.04	2.36
k_1_	0.47	0.78	0.37	0.08	0.72	0.59	0.1	1.24	0.58	0.58	1.03	0.41	0.80	1.48	0.78
R^2^	0.99	0.89	0.99	0.98	0.98	0.99	0.99	0.99	0.99	0.97	0.93	0.91	0.99	0.99	0.97
Δq, %	4.51	6.71	4.73	70.9	0.83	4.21	57.1	2.15	2.02	3.21	3.33	2.62	1.13	2.28	1.89
**PSO**															
q_max_	6.11	5.77	9.98	16.87	5.14	6.18	11.7	3.58	4.79	3.01	2.74	3.69	1.92	2.22	2.81
k_2_	0.07	0.49	0.03	0.01	0.16	0.10	0.01	0.53	0.14	0.18	0.81	0.25	0.5	1.21	0.34
R^2^	0.99	0.99	0.99	0.98	0.99	0.99	0.99	0.99	0.99	0.97	0.99	0.99	0.99	0.99	0.98
Δq, %	20.8	4.71	24.4	170.8	8.54	15.9	135.	2.89	12.3	17.8	4.18	8.42	8.15	2.89	6.2
**Elovich**															
α	4.54	9.98	6.19	2.05	9.99	7.57	1.86	8.09	6.50	2.83	13.0	8.76	4.73	16	7.24
β	0.66	0.59	0.43	0.57	0.92	0.70	0.79	1.22	0.93	1.32	1.95	1.34	2.57	2.62	1.79
R^2^	0.99	0.99	0.99	0.99	0.99	0.98	0.97	0.98	0.99	0.96	0.97	0.99	0.99	0.98	0.99
Δq, %	2.13	2.39	0.39	0.26	0.30	3.09	2.50	2.05	1.30	4.07	2.85	1.1	0.40	1.01	0.16
**DI**															
k	1.84	2.67	2.69	1.29	1.95	2.16	1.08	1.59	1.70	1.00	1.21	1.44	0.75	1.02	1.1
R^2^	0.97	0.97	0.99	0.99	0.95	0.95	0.93	0.94	0.97	0.92	0.95	0.93	0.92	0.95	0.94
Δq, %	3.51	8.77	0.60	4.53	4.62	7.34	4.28	6.95	4.77	6.22	8.54	5.95	5.99	8.88	4.24
**DE**															
k_Ext_	0.07	0.12	0.12	0.11	0.22	0.25	0.15	0.32	0.36	0.20	0.34	0.50	0.22	0.40	0.44
R^2^	0.93	0.95	0.93	0.98	0.93	0.94	0.99	0.92	0.95	0.87	0.92	0.94	0.91	0.89	0.92
Δq, %	9.43	13.7	5.35	0.94	7.69	25.3	28.2	8.75	5.17	9.65	8.74	5.77	26.5	10.7	3.29

**Table 6 molecules-26-06918-t006:** Kinetic constant of DNS adsorption patterns in GAC.

Model	2 g/L			4 g/L			6 g/L			8 g/L			10 g/L		
PFO	25 °C	35 °C	45 °C	25 °C	35 °C	45 °C	25 °C	35 °C	45 °C	25 °C	35 °C	45 °C	25 °C	35 °C	45 °C
q_max_	7.0	9.14	10.3	5.84	6.62	6.50	5.25	4.04	4.26	4.79	3.14	3.37	4.0	3.0	2.65
k_1_	1.83	2.32	2.74	3.20	3.10	3.87	2.36	2.94	4.40	1.72	2.87	2.354	2.11	2.46	2.52
R^2^	0.99	0.99	0.99	0.99	0.99	0.99	0.99	0.99	0.99	0.99	0.99	0.97	0.99	0.99	0.98
Δq, %	4.49	1.46	1.66	2.15	1.11	0.51	1.42	1.55	0.42	8.52	1.32	2.68	2.49	7.63	1.86
**PSO**															
q_max_	7.58	9.56	10.7	5.97	6.79	6.45	5.46	4.16	4.3	4.65	3.23	3.18	4.21	2.54	2.54
k_2_	0.44	0.32	0.78	2.27	1.79	9.45	1.25	2.41	9.13	2.56	3.15	9.33	1.22	4.13	2.54
R^2^	0.99	0.99	0.99	0.99	0.99	0.99	0.99	0.99	0.99	0.98	0.99	0.99	0.99	0.99	0.99
Δq, %	1.14	0.54	0.13	1.19	0.03	0.13	0.30	0.24	0.05	0.98	0.06	0.02	0.27	0.52	0.02
**Elovich**															
α	9.53	8.97	10.5	10.7	9.44	11.0	11.8	12.9	12.0	12.1	13.0	13.5	12.9	13.9	14.9
β	0.42	0.28	0.26	0.54	0.45	0.52	0.63	0.90	0.90	0.73	1.26	1.34	0.92	1.73	1.79
R^2^	0.83	0.69	0.70	0.77	0.75	0.76	0.84	0.86	0.84	0.90	0.88	0.87	0.90	0.90	0.89
Δq, %	1.47	10.4	0.21	3.14	2.54	8.54	4.85	15.9	3.46	3.17	5.57	14.7	4.13	5.39	4.98
**DI**															
k	3.55	4.68	5.33	3.01	3.41	3.3	2.69	2.08	2.2	2.42	1.67	1.62	2.04	1.27	1.29
R^2^	0.73	0.67	0.64	0.60	0.60	0.53	0.64	0.62	0.55	0.72	0.62	0.56	0.68	0.62	0.56
Δq, %	5.34	9.59	9.59	9.02	10.3	11.6	9.58	9.77	11.3	3.01	9.99	11.1	8.11	9.50	11.0
**DE**															
k_Ext_	0.18	0.31	0.47	0.97	0.69	0.73	1.20	1.16	1.0	1.27	1.04	1.54	2.54	1.29	1.66
R^2^	0.38	0.49	0.60	0.77	0.76	0.75	0.83	0.85	0.87	0.48	0.89	0.90	0.45	0.87	0.90
Δq, %	9.29	18.8	10.8	6.82	1.52	3.95	5.95	5.74	4.27	28.7	4.23	65.1	14.6	4.70	5.11

**Table 7 molecules-26-06918-t007:** Physicochemical properties and elemental analysis of PAC [45,46].

PZC	S_a_ (Superficial Area), m^2^/g	Vp (Pore Volumen), cm^3^/g	Dp (Pore Diameter), nm	Elemental Composition, %*w/w*				
				N	C	H	S	O
7.2	650	0.94	41.1	0.43	79.1	2.43	1.03	5.67

**Table 8 molecules-26-06918-t008:** Surface acid groups in PAC.

Carboxylic Groups, mol/nm^2^	Laconic Groups, mol/nm^2^	Phenolic Groups, mol/nm^2^	Total Acidity, mol/nm^2^
0.7079	0.7638	1.7899	3.2616

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
