# Peer review of "3,5-Dinitrosalicylic Acid Adsorption Using Granulated and Powdered Activated Carbons"

_molecules, 2021, doi:10.3390/molecules26226918_

Round 1

Reviewer 1 Report

The authors report the adsorption behavior of DNS on commercial activated carbon in granular and powdered form, at different temperatures. The results show that the adsorption takes place in more than one layer in equilibrium and is favorable for the removal of DNS in both GAC and PAC. The mass transfer and absorption kinetic are investigated detailly in the manuscript. Two different removal mechanism were proposed by authors for DNS absorption on GAC and PAC respectively based on experimental results. In general, this study provides valuable data for the DNS absorption, and this could be helpful for removing them from wastewater in the future. I recommend to accept the manuscript in present form. 

Author Response

The reviewer was very kind to take into account the effort that was made to carry out this work, with which we appreciate each of his words and this encourage us to making a greater effort to continue contributing to knowledge.

Reviewer 2 Report

The paper presents research on 3,5-dinitrosalysilic acid adsorption using granulated and powdered activated carbon. The article contains substantive and methodological errors. Additional experimental measurements are required. The presentation of methods and scientific results in the current form is unsatisfactory for publication in the Molecules journal. The minor and significant drawbacks to be addressed can be specified as follows:
1.    The title must be changed. (1) 3, 5 Dinitrosalysilic ---> 3,5-Dinitrosalysilic Acid. (2) Granulated (GAC) and Powdered Activated Carbon (PAC) ---> Granulated and Powdered Activated Carbons.
2.    Fig. 1. (1) I suggest introducing colours. See Fig.3. (2) Why did the authors add lines to the points? What do they mean? Spline? If they are not related to the data description theoretical models, I suggest removing them!
3.    Tab. 1. (1) The values of the adjusted coefficient of determination should be considered. Classical R2 shows how well terms (data points) fit a curve or line. Adjusted R2 also indicates how well terms fit a curve or line, but adjusts for the number of terms in a model (for example DI - one best fit parameter/term and Elevich – two parameter/term). The comparison of Adjusted R2 allows only to compare different models (with different variables). The next problem, the results obtained indicate compliance with the rule - the more best fit parameters, the better the compatibility between experimental and theoretical data. (2) The values are given with too many decimal places. 
4.    Tab. 2. (1) 0.331I strange vertical dash line. (2) Why is the deltaG value twice as high for GAC compared to PAC?
5.    Fig. 3. Why did the authors add lines to the points? What do they mean? Spline? If they are not related to the data description theoretical models, I suggest removing them!
6.    Tabs. 3 and 4. See my comments to Tab. 1.
7.    Fig. 4. (a) The straight-line description of the data over the entire range seems highly questionable (Fig. 4a, GAC). (b) Why was the same data scale not used on the y- axis (everywhere up to 3.5)?
8.    Page 10. Text and Tabs. 5 and 6. AC ---? GAC? PAC? Chaos – I'm confused. I'm sorry. “Superficial area”, Vp, and Dp ??
9.    Fig. 6. Data for GAC?
10.    Pages 11 and 12. No information on PAC! If this is information about PAC "Part of GAC 304 was pulverized with a ball mill (Gunt Hamburg CE-245) and sieved (Ro-Tap RX-30) to a 305 particle size of 100 mesh (0.297 mm)?" how did this treatment affect the chemical nature of the PAC surface and its porous structure. From the analysis of adsorption isotherms, it can be seen that the adsorption process differs drastically. It is necessary to measure nitrogen adsorption (T=77.5 K)and XPS
11.    Eq. 6. Where was RDNS used?
12.    There is no comparison of the sorption capacity of these materials with data (worse/better) taken from the literature for other DNS/adsorbent systems.

Author Response

Dear reviewer, thank you very much for your comments and observations that allow me to better and enrich this work. Based on your analysis, allow me to mention the following:

  1. Consideration was taken to change the title to better reflect the work done on the manuscript.
  2. To avoid confusion the colors of the experimental data were changed and the trend line of these data was removed as proposed.
  3. In table 1, the significant figures were reduced as requested and regarding the use of the deterministic coefficient (R2) it was used in the first instance to choose the best models of the 6 that were proposed to describe the experimental data obtained, without However, the normalized standard deviation (Δq) was used for which model appropriately describes the adsorption isotherm, since if this value is very small, it indicates that the experimental value with respect to the theoretical data of the maximum adsorption capacity is very close.
  4. The error you mark was corrected and with respect to your question about ΔG, I think it refers more to ΔH where it is effectively almost double for GAC compared to PAC, this is because more energy is needed to remove DNS of the solution as also shown in the calculation of the removal energy in the DR model.
  5. With respect to the other tables, the decimals of the data were also reduced.
  6. The colors of the experimental data were changed and the trend line of these data was removed as proposed to avoid confusion.
  7. The model to determine the limitations by mass transfer does not imply that they must have the same range but to what extent the linearity of the process exists, since if it is subjected to the same range of time indicates that it does not matter there is an effect due to the presence of the adsorbent concentration in the solution,
  8. In this paragraph and tables it is specified what type of adsorbent it refers to (PAC)
  9. It was decided to do the FTIR studies, as well as the textural properties only for PAC, since it is the best adsorbent than GAC.
  10. The study of the acid groups was carried out on the adsorbent with the highest adsorption capacity (for economy), so the GAC data is not available, only for PAC.
  11. Figure 2 shows the use of the removal percentage calculation (%RDNS).
  12. The request was specified, however, throughout the manuscript the comparison with several pollutants of a similar nature to the DNS is mentioned (pages 4, 5, 6 and 8), in addition there are only 2 references (14 and 21) in the literature that directly mentions the treatment of this pollutant.

Reviewer 3 Report

The paper entitled “3, 5 Dinitrosalysilic (DNS) Adsorption using Granulated (GAC) and Powdered Activated Carbon (PAC)” reported an interesting work on the very important topic of the contamination of water.

The article is well written, aimed to study the adsorption of DNS in equilibrium and dynamically using GAC and PAC at different temperatures, determining if the process has limitations due to external and intraparticle mass transfer.  The manuscript needs several revisions before it can be published. Therefore, please improved/clarified the following points:

  1. Please correct in the title Dinitrosalysilic with Dinitrosalycilic. I recommend to define DNS, GAC and PAC as first appear in the abstract as you have already done, and remove it from the title.
  2. The adsorption needs to be conducted at different pH in order to get the maximum adsorption capacity and to optimize all the parameters.
  3. For how many cycles GAC and PAC can be used in case of DNS adsorption.
  4. How the desorption is performed?
  5. As a minor point, check again the use of subscript and superscript (ex. Page 12 line 345 “RL”)

Author Response

Dear reviewer, thank you very much for your observations, comments and suggestions to improve and enrich this work.  I commented to you the following:

1. The misspelled word in the title of the work was changed, as well as following your recommendations with the other three words
2. In effect, the change in pH can favor the adsorption of pollutants, however for our case study it was decided to leave the pH of the original solution, because this value is the one that the solution has when it is discarded after tests for reducing sugars in wines and sugary drinks.
3. In the case of activated carbon in its two presentations, it was observed that it can be reused more than 6 times, without significantly modifying its adsorption capacity.
 4. If wants to recover the pollutant from the activated carbon, it is complicated and expensive since you need a recovery unit that mainly consists of leaching the component of interest.
5. The entire document was reviewed and all errors in the manuscript were corrected.

Round 2

Reviewer 2 Report

Congratulations on a great job. The author has made a substantial improvement for this article. The manuscript can be accepted for publishment in the present form.
Unfortunately, there are still a lot of annoying bugs/errors/typos at work. I suggest reviewing the work that I believe is underdeveloped. Of course, I leave the decisions to the editor.
For example – Tab. 5.
a)    Superficial area – Specific area
b)    pore diameter - pore diameter
c)    volumen – volume
d)    value of mean pore width, i.e. 41.1 nm seems very strange. I tried to find it in the Internet resources, but I failed. (https://www.reskem.com/wp-uploads/2015/04/norit-darco2040.pdf).

Author Response

Dear reviewer, thank you very much for your comments, since they made the manuscript have a significant improvement, making the modifications and above all, exhaustively reviewing the work to correct the errors that were found.

Reviewer 3 Report

After revision the paper was significantly improved and I consider it suitable for publication in its present form.

Author Response

Dear reviewer, thank you very much for your comments and suggestions to enrich the manuscript.